# WISP-3 Stimulates VEGF-C-Dependent Lymphangiogenesis in Human Chondrosarcoma Cells by Inhibiting miR-196a-3p Synthesis

**DOI:** 10.3390/biomedicines9101330

**Published:** 2021-09-27

**Authors:** Chih-Yang Lin, Shih-Wei Wang, Jeng-Hung Guo, Huai-Ching Tai, Wen-Chun Sun, Cheng-Ta Lai, Chen-Yu Yang, Shih-Chia Liu, Yi-Chin Fong, Chih-Hsin Tang

**Affiliations:** 1Department of Pharmacology, School of Medicine, China Medical University, Taichung 404022, Taiwan; u9957651@cmu.edu.tw; 2Institute of Biomedical Sciences, Mackay Medical College, Taipei 104217, Taiwan; shihwei@mmc.edu.tw (S.-W.W.); sunwenchin@gmail.com (W.-C.S.); 3Department of Medicine, Mackay Medical College, New Taipei City 252005, Taiwan; laichengta@gmail.com; 4Graduate Institute of Natural Products, College of Pharmacy, Kaohsiung Medical University, Kaohsiung 807378, Taiwan; 5Graduate Institute of Biomedical Sciences, China Medical University, Taichung 404022, Taiwan; muttguo@gmail.com; 6Department of Neurosurgery, China Medical University Hospital, Taichung 404022, Taiwan; 7School of Medicine, Fu-Jen Catholic University, New Taipei City 252005, Taiwan; taihuai48@hotmail.com; 8Department of Urology, Fu-Jen Catholic University Hospital, New Taipei City 252005, Taiwan; 9Department of Surgery, Division of Colon and Rectal Surgery, MacKay Memorial Hospital, Taipei 104217, Taiwan; 10Department of Orthopedic Surgery, Division of Pediatric Orthopedics, MacKay Memorial Hospital, Taipei 104217, Taiwan; meinen1018@hotmail.com (C.-Y.Y.); jasonscliu649@gmail.com (S.-C.L.); 11Department of Sports Medicine, College of Health Care, China Medical University, Taichung 404022, Taiwan; 12Department of Orthopedic Surgery, China Medical University Beigang Hospital, Yunlin 651012, Taiwan; 13Department of Biotechnology, College of Health Science, Asia University, Taichung 413005, Taiwan; 14Chinese Medicine Research Center, China Medical University, Taichung 404022, Taiwan

**Keywords:** WISP-3, chondrosarcoma, VEGF-C, lymphangiogenesis, miR-196a-3p

## Abstract

Chondrosarcoma is a malignant bone tumor with high metastatic potential. Lymphangiogenesis is a critical biological step in cancer metastasis. WNT1-inducible signaling pathway protein 3 (WISP-3) regulates angiogenesis and facilitates chondrosarcoma metastasis, but the role of WISP-3 in chondrosarcoma lymphangiogenesis is unclear. In this study, incubation of chondrosarcoma cells with WISP-3 increased the production of VEGF-C, an important lymphangiogenic factor. Conditioned medium from WISP-3-treated chondrosarcoma cells significantly enhanced lymphatic endothelial cell tube formation. WISP-3-induced stimulation of VEGF-C-dependent lymphangiogenesis inhibited miR-196a-3p synthesis in the ERK, JNK, and p38 signaling pathways. This evidence suggests that the WISP-3/VEGF-C axis is worth targeting in the treatment of lymphangiogenesis in human chondrosarcoma.

## 1. Introduction

Chondrosarcomas are cartilage-forming tumors found typically in the femur, tibia, or pelvis [1,2], and these tumors easily metastasize to distant organs [1]. In particular, high-grade chondrosarcomas are prone to metastasize to the lungs [3,4] and lack effective therapeutic options [5], so it is imperative that research efforts search for potentially effective treatments.

Tumor metastasis involves the movement of cancer cells from the primary site and their establishment in other organs [6,7]. A growing body of research has highlighted the role of lymphangiogenesis in cancer metastasis [6,8]. Lymphangiogenesis enables lymphatic endothelial cells (LECs) to proliferate and migrate through lymphatic vessels surrounding the tumors [9,10]. Vascular endothelial growth factor (VEGF)-C is a key factor in the regulation of lymphangiogenesis [9,10], as increasingly higher levels of VEGF-C expression stimulate LEC-associated lymphangiogenesis and the metastatic potential of chondrosarcoma cells, while VEGF-C expression is significantly higher in human chondrosarcoma tissue than in normal cartilage [11,12]. Thus, it is critical to investigate the mechanism of VEGF-C synthesis in human chondrosarcoma cells.

CCN (Cyr61, CTGF, and Nov) family proteins are important for tumor development and metastasis [2,13]. One CCN family member, WNT1-inducible signaling pathway protein 3 (WISP-3, also known as CCN6), regulates several cellular functions [14]. WISP-3 expression has been detected in different types of cancers [15,16]. We have previously found higher WISP-3 levels in human chondrosarcoma tissue than in normal cartilage [17]. Moreover, reports of WISP-3-induced promotion of VEGF-A production and cellular motility in human chondrosarcoma cells reinforces the data, showing that WISP-3 mediates angiogenesis and metastasis in chondrosarcoma [17,18]. Thus, WISP-3 appears to be a novel avenue for treating metastatic chondrosarcoma.

MiRNAs, single-stranded noncoding RNA molecules that manipulate gene expression at the post-transcriptional level [19], have the ability to regulate inflammatory and immune responses [20], and negatively or positively affect the proliferation, differentiation, migration, and survival of cancer cells [21,22]. Moreover, miRNAs have been found to regulate lymphangiogenic activity during cancer progression [23]. However, it remains unclear as to whether the WISP-3-miRNA axis regulates lymphangiogenesis in chondrosarcoma. Our study has identified that WISP-3 increased VEGF-C production and facilitated LEC lymphangiogenesis in human chondrosarcoma cells by inhibiting miR-196a-3p synthesis in the ERK, JNK, and p38 signaling pathways. Inhibition of WISP-3 expression reduced VEGF-C-dependent lymphangiogenesis in vivo. WISP-3 therefore seems to be a novel therapeutic target for chondrosarcoma.

## 2. Materials and Methods

### 2.1. Materials

WISP-3, VEGF-C, ERK, p38, JNK, and β-actin antibodies were obtained from GeneTex (Hsinchu, Taiwan). The phosphorylated forms of ERK, p38, and JNK antibodies were bought from Cell Signaling (Danvers, MA, USA). ERK, p38, JNK, and control ON-TARGETplus siRNAs were purchased from Dharmacon (Lafayette, CO, USA). Taqman^®^ One-Step PCR Master Mix and qPCR primers and probes were bought from Applied Biosystems (Foster City, CA, USA). Recombinant human WISP-3 was acquired from PeproTech (Rocky Hill, NJ, USA). All other chemicals not already mentioned were acquired from Sigma-Aldrich (St. Louis, MO, USA).

### 2.2. Cell Culture

Human LECs were purchased from Lonza (Walkersville, MD, USA) and cultured in EGM-2 MV medium consisting of EBM-2 basal medium and SingleQuots Kit media. The human chondrosarcoma cell line SW1353 was bought from ATCC (Manassas, VA, USA), and cultured in Dulbecco’s Modified Eagle Medium (DMEM). The chondrosarcoma JJ012 cell line was kindly provided by Dr. Sean P. Scully (University of Miami School of Medicine, Miami, FL, USA). Highly migratory JJ012(S10) cells and stable JJ012(S10) cells infected with WISP-3 shRNA were established as per the methods detailed in our previous report [18]. JJ012 cell lines were cultured 50%/50% in DMEM/α-MEM. For all three chondrosarcoma cell lines, the culture also included 10% fetal bovine serum, 1% streptomycin (100 μg/mL), sodium bicarbonate (1.5 g/L), sodium pyruvate (1 mM), and HEPES (25 mM) at 37 °C in 5% CO_2_.

### 2.3. Western Blot

After the indicated treatments, chondrosarcoma cells were lysed in RIPA buffer containing phosphatase and protease inhibitors. The protein lysates were separated using SDS-PAGE then transferred to PVDF membranes, as described in our previous publications [24,25,26]. Membranes were blocked for 1 h with TBST (TBS with 0.1% Tween 20) containing 4% non-fat milk, then washed three times in TBST buffer before administration of antibodies targeting p-ERK, p-JNK, p-p38, WISP-3, VEGF-C, and β-actin for 1 h, then washed again (three times) with TBST buffer. The membranes were then incubated for 1 h with HRP-conjugated secondary antibodies, and subsequently washed three times with TBST buffer. Immunoreactive signals were visualized by enhanced chemiluminescence with ECL reagent and blot membranes were visualized with a Fujifilm LAS-4000 imaging system (GE Healthcare, Little Chalfont, UK).

### 2.4. mRNA and miRNA Quantification

A TRIzol kit (MDBio, Taipei, Taiwan) extracted total RNA and miRNA from the chondrosarcoma cell lines, according to the manufacturer’s instructions, then examined them with a NanoVue Plus spectrophotometer (GE Healthcare Life Sciences; Pittsburgh, PA, USA). Following the manufacturers’ instructions, we reverse-transcribed total RNA into complementary DNA (cDNA) using the M-MLV RT kit (Thermo Fisher Scientific; Waltham, MA, USA) and the Mir-X™ miRNA First-Strand Synthesis kit (Terra Bella Avenue; Mountain View, CA, USA). Quantitative real-time PCR (qRT-PCR) was performed using the miR-196a-3p specific primer. U6 was used as a normalizing control for miRNA qRT-PCR analysis. cDNA samples were subjected to qRT-PCR analysis with SYBR Green, as per our previous reports [27,28].

### 2.5. Collection of Chondrosarcoma Conditioned Medium and the ELISA Assay

Chondrosarcoma cells were cultured and grown to confluence. The culture medium was then exchanged with serum-free DMEM/α-MEM medium. Cells were pretreated or transfected with the indicated inhibitors or siRNAs then stimulated with WISP-3 for 24 h. The medium was collected as conditioned medium (CM) and stored at −80 °C until use. Secreted VEGF-C was analyzed by a VEGF-C ELISA assay kit, according to the manufacturer’s instructions [29].

### 2.6. LEC Tube Formation

LECs were suspended at a density of 3 × 10^5^ (50% EGM-2MV medium and 50% chondrosarcoma cell CM) and cultured in 48-well plates precoated with 150 μL of Matrigel. LEC tube formation was photographed after 6 h and the number of tube branches was counted manually [29].

### 2.7. Tumor Xenograft Study

JJ012, JJ012(S10), or JJ012(S10)/WISP-3-shRNA cells (5 × 10^6^) were transplanted subcutaneously into the right flanks of BALB/c-nu mice (4-week-old males) according to a previous protocol [18]. After 4 weeks, the mice were sacrificed by CO_2_ inhalation and the tumors were removed. All animal work was carried out in accordance with a protocol approved by China Medical University (Taichung, Taiwan) Institutional Animal Care and Use Committees (CMUIACUC-2017-151-1).

### 2.8. Immunohistochemistry (IHC) Staining

Mouse tumor tissues were rehydrated and treated with primary anti-VEGF-C or LYVE-1 antibodies, then incubated with biotin-labeled secondary antibody. The slides were treated with the ABC Kit (Vector Laboratories, Burlingame, CA, USA) according to a previous protocol [30], and then photographed using the microscope. Intra-tumoral lymph vessels were defined as LYVE-1-positive (LYVE-1^+^) vessels that were in close contact with tumor cells or located in the desmoplastic stroma. Peritumoral lymph vessels were defined as LYVE-1^+^ vessels at a maximum distance of 2 mm from the tumor periphery. In each case, five microscopic fields “lymphatic vessel hot spots” were examined at high power. The mean of these five values was recorded as the LYVE-1^+^. Density measurements of lymph vessels’ area were performed with ImageJ software [31,32].

### 2.9. Statistical Analysis

All values are presented as the mean ± standard deviation (SD). Differences between two experimental groups were assessed for significance using the Student’s *t*-test and considered to be significant if the *p*-value was <0.05.

## 3. Results

### 3.1. WISP-3 Facilitates VEGF-C-Dependent Lymphangiogenesis in Chondrosarcoma Cells

WISP-3 facilitates angiogenesis and metastasis in human chondrosarcoma cells [17,18] and VEGF-C reportedly regulates lymphangiogenesis in different cancer cells [33]. We initially found that WISP-3 treatment increased VEGF-C mRNA and secreted protein production in JJ012 and SW1353 cells, according to qPCR and ELISA data (Figure 1A,B). LEC tube formation is a well-established model that is used to mimic lymphangiogenesis in vitro [10]. CM from WISP-3-treated chondrosarcoma cells significantly enhanced tube formation in LECs (Figure 1C,D). VEGF-C mAb, but not control IgG, reduced the effects of WISP-3 on VEGF-C-promoted LEC tube formation (Figure 1C,D), which suggests that WISP-3 induces chondrosarcoma lymphangiogenesis in a VEGF-C-dependent manner.

### 3.2. The MAPK Signaling Pathway Mediates the Effect of WISP-3 upon VEGF-C Synthesis of Human Chondrosarcoma Cells

The MAPK (ERK, JNK, and p38) signaling pathway is important in the metastatic process of chondrosarcoma [34,35]. Stimulation of chondrosarcoma cell lines with ERK, JNK, or p38 inhibitors (ERK II, SP600125, or SB203580, respectively) significantly reduced WISP-3-enhanced stimulation of VEGF-C production (Figure 2A,C, Figure 3A,C and Figure 4A,C). Similar effects were observed when the chondrosarcoma cell lines were transfected with ERK, JNK, or p38 siRNAs (Figure 2B,D, Figure 3B,D and Figure 4B,D), which substantially inhibited ERK, JNK, or p38 expression, respectively (Figure 2F, Figure 3F and Figure 4F). WISP-3 stimulation time-dependently promoted ERK, JNK, and p38 phosphorylation. These results suggest that the MAPK signaling pathway regulates WISP-3-promoted VEGF-C expression and lymphangiogenesis.

### 3.3. Inhibition of miR-196a-3p Controls WISP-3-Promoted VEGF-C Synthesis

MiRNA expression is dysregulated in cancer patients and differs from miRNA expression in healthy individuals [36,37]. Using open-source miRNA software, we identified six miRNAs that potentially target with VEGF-C transcription, and miR-196-3p synthesis was mostly decreased with WISP-3 treatment (Figure 5A). WISP-3 concentration-dependently reduced miR-196a-3p and precursor miR-196a-3p (pre-miR-196a-3p) synthesis in the chondrosarcoma cells (Figure 5B,C). Transfection of chondrosarcoma cells with miR-196a-3p mimic antagonized the effects of WISP-3 upon VEGF-C production (Figure 5D). Treatment of cells with ERK, JNK, and p38 inhibitors or siRNAs all reversed WISP-3-induced inhibition of miR-196a-5p expression (Figure 5E,F), indicating that MAPK signaling mediates WISP-3-induced inhibition of miR-196a-3p.

### 3.4. Inhibition of WISP-3 Reduces LEC Lymphangiogenesis In Vivo

We previously established highly migratory JJ012(S10) cells by using Transwell [18,38]. Here, we found that JJ012(S10) cells expressed higher protein levels of WISP-3 and VEGF-C, and that CM from the JJ012(S10) cell line strongly promoted LEC tube formation compared to parental JJ012 cells (Figure 6A–D). Knockdown of WISP-3 in JJ012(S10) cells by WISP-3 shRNA inhibited WISP-3 and VEGF-C expression, and also reduced LEC tube formation (Figure 6A–D). In the tumor-induced lymphangiogenesis model, the IHC data revealed that knockdown of WISP-3 inhibited chondrosarcoma-promoted expression of WISP-3 and LEC markers VEGF-C and LYVE-1 (Figure 6E–I). These results indicate that inhibiting WISP-3 lowers LEC lymphangiogenesis in vivo.

## 4. Discussion

Chondrosarcoma is responsible for around one-fourth (~26%) of all bone cancers [39] and is well-characterized as being an aggressive malignancy with a high likelihood of metastasis [40]. Up until now, chondrosarcoma metastasis lacks effective adjuvant therapies [1,3]. Lymphangiogenesis is a critical step during tumor metastasis, promoting tumor development via the synthesis of lymphatic vessels [10]. Evidence suggests that increasing levels of VEGF-C expression are associated with tumor relapse and a poor prognosis [41,42]. Thus, VEGF-C represents a critical candidate for preventing lymphangiogenesis and metastasis [41,42]. Our cellular and preclinical investigations found that WISP-3 reliably promotes VEGF-C-dependent lymphangiogenesis in chondrosarcoma. We confirmed that WISP-3 facilitates VEGF-C production in chondrosarcoma and subsequently increases LEC lymphangiogenesis by inhibiting miR-196a-5p expression in the MAPK signaling pathway.

WISP-3 protein is involved in the development, homeostasis, and repair of mesenchymal tissues [43]. WISP-3 dose-dependently stimulates the migration of undifferentiated mesenchymal stem cells (MSCs), and the chemotactic activity of WISP-3 in these cells is reportedly mediated by integrin ανß5 [43]. WISP-3 also has effects on cartilage homeostasis. For example, WISP-3 overexpression in normal cartilage (C-28/I2) cells dramatically reduces the expression of ADAMTS-4 and ADAMTS-5, and markedly elevates matrix metalloproteinase-1 (MMP-1) and MMP-10 expression, leading to the degradation of cartilage and the development of osteoarthritis [44]. Thus, WISP-3 plays a critical role in influencing the development of bone disease, and by the recruitment of MSCs by stimulating their migration in skeletal development and repair.

LECs stimulate lymphatic vessel formation, and the promotion of LEC mobilization by lymphangiogenic factors facilitates tumor development and angiogenesis [45]. Here, we observed that CM from highly migratory JJ012(S10) cells easily facilitated LEC tube formation compared to parental JJ012 cells. WISP-3 shRNA reduced VEGF-C production, LEC tube formation, and the expression of LEC markers in vivo. Thus, inhibiting WISP-3 inhibits LECs lymphangiogenesis in vitro and in vivo. Various lymphangiogenic factors, including VEGF-A, VEGF-C, and VEGF-D, are involved in the lymphangiogenic processes of several different diseases, including cancer [9,10,46]. We have previously reported that WISP-3 promotes VEGF-A expression in chondrosarcoma cells and induces endothelial progenitor cell angiogenesis [18]. In addition, an in vivo tumor xenograft study reveals that inhibiting WISP-3 expression reduces the expression of the vessel markers CD31 and VEGF-A, indicating that WISP-3 enhances angiogenesis in vivo [18]. Here, we report that incubation of chondrosarcoma cell lines with WISP-3 concentration-dependently promotes mRNA and VEGF-C synthesis, resulting in LEC lymphangiogenesis. The VEGF-C antibody abolished LEC tube formation in CM from WISP-3-treated chondrosarcoma cells. Moreover, we also used a mouse tumor xenograft model to simulate the environment of initial lymphangiogenesis. Our results revealed that knockdown of WISP-3 inhibited chondrosarcoma-promoted expression of the LEC markers VEGF-C and LYVE-1, indicating that VEGF-C is a critical factor in WISP-3-induced lymphangiogenesis in human chondrosarcoma. Whether other lymphangiogenic factors also regulate WISP-3-enhanced promotion of lymphangiogenesis in chondrosarcoma needs further investigation.

Activation of the MAPK pathway is important in the adjustment of different cellular effects [47,48]. This signaling pathway regulates the expression of VEGF-C-associated cellular functions [49,50]. Our results show that WISP-3 increases the phosphorylation of ERK, JNK, and p38, while their respective pharmacological inhibitors suppress WISP-3-induced promotion of VEGF-C expression. This phenomenon is confirmed by similar effects observed with genetic siRNAs of ERK, JNK, and p38. This evidence reveals that activation of ERK, JNK, and p38 signaling controls WISP-3-enhanced promotion of VEGF-C synthesis and lymphangiogenesis of chondrosarcoma cells.

MiRNAs post-transcriptionally regulate gene expression [51]. In tumors, aberrant miRNA expression regulates the expression of lymphangiogenic pathways [12]. Numerous miRNAs also control lymphangiogenesis during tumor progression [29]. In this study, stimulation of chondrosarcoma cells with WISP-3 inhibited miR-196a-5p expression, and transfecting them with miR-196a-5p mimic antagonized WISP-3-promoted upregulation of VEGF-C expression and LEC lymphangiogenesis. Treating the chondrosarcoma cells with MAPK inhibitors or siRNAs reversed WISP-3-promoted inhibition of miR-196a-5p expression, suggesting that WISP-3 may assist with VEGF-C production and LEC lymphangiogenesis by inhibiting miR-196a-5p synthesis via the MAPK signaling cascades.

## 5. Conclusions

In conclusion, our study has identified that WISP-3 facilitates VEGF-C-dependent lymphangiogenesis of chondrosarcoma cells by inhibiting miR-196a-5p synthesis in the ERK, JNK, and p38 pathways (Figure 7). We believe that targeting WISP-3-dependent VEGF-C expression in metastatic chondrosarcoma offers a new way to address this aggressive malignancy.

## Figures and Tables

**Figure 1 biomedicines-09-01330-f001:**
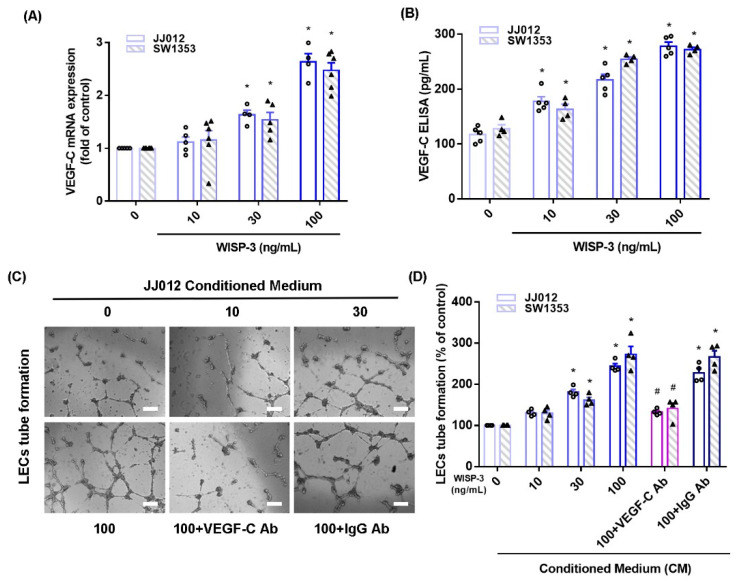
**WISP-3 promotes VEGF-C-dependent lymphangiogenesis in human chondrosarcoma**. (**A**,**B**) Cells were incubated with WISP-3 (10–100 ng/mL) and levels of VEGF-C mRNA and protein expression were examined by qPCR and ELISA assays. (**C**,**D**) Collected conditioned medium (CM) was applied to lymphatic endothelial cells (LECs), then LEC tube formation was measured (scale bar = 100 μm). Quantitative results are expressed as the mean ± SD. All experiments were repeated 3 to 5 times (solid squares, solid dots, hollow circles and solid triangles). * *p* < 0.05 compared with the control group; # *p* < 0.05 compared with the WISP-3-treated group.

**Figure 2 biomedicines-09-01330-f002:**
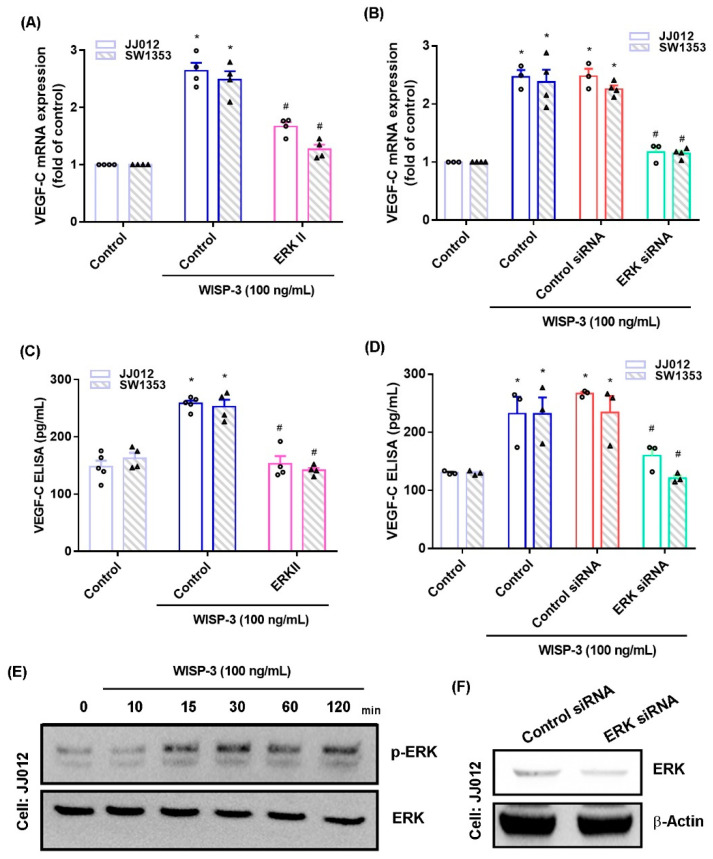
**The ERK****pathway controls WISP-3****-induced VEGF-C production.** (**A**–**D**) Cells were pretreated with a ERK inhibitor (ERK II) or transfected with an ERK siRNA, then stimulated with WISP-3. Levels of VEGF-C expression were examined by qPCR and ELISA. (**E**) JJ012 cells were incubated with WISP-3 for the indicated time intervals, and ERK phosphorylation was examined by Western blot. (**F**) JJ012 cells were transfected with ERK siRNA, and ERK expression was examined by Western blot. Quantitative results are expressed as the mean ± SD. All experiments were repeated 3 to 5 times (solid squares, solid dots, hollow circles and solid triangles). * *p* < 0.05 compared with the control group; # *p* < 0.05 compared with the WISP-3-treated group.

**Figure 3 biomedicines-09-01330-f003:**
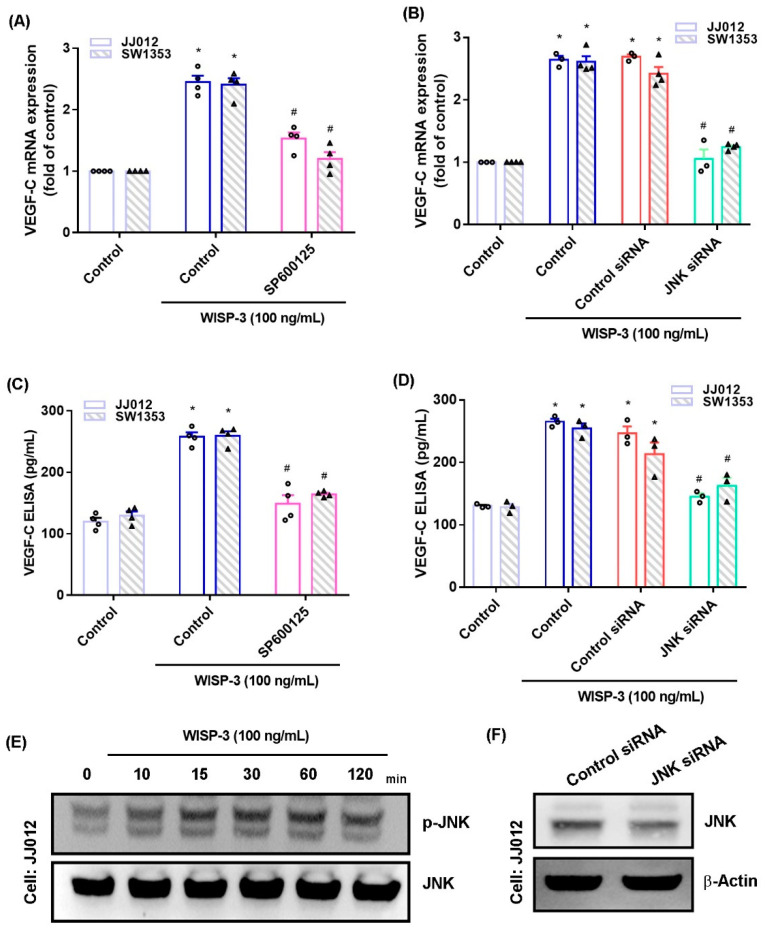
**The JNK****pathway controls WISP-3****-induced VEGF-C production.** (**A**–**D**) Cells were pretreated with a JNK inhibitor (SP600125) or transfected with a JNK siRNA, then stimulated with WISP-3. The levels of VEGF-C expression were examined by qPCR and ELISA. (**E**) JJ012 cells were incubated with WISP-3 for the indicated time intervals, and JNK phosphorylation was examined by Western blot. (**F**) JJ012 cells were transfected with JNK siRNA, and JNK expression was examined by Western blot. Quantitative results are expressed as the mean ± SD. All experiments were repeated 3 to 5 times (solid squares, solid dots, hollow circles and solid triangles). * *p* < 0.05 compared with the control group; # *p* < 0.05 compared with the WISP-3-treated group.

**Figure 4 biomedicines-09-01330-f004:**
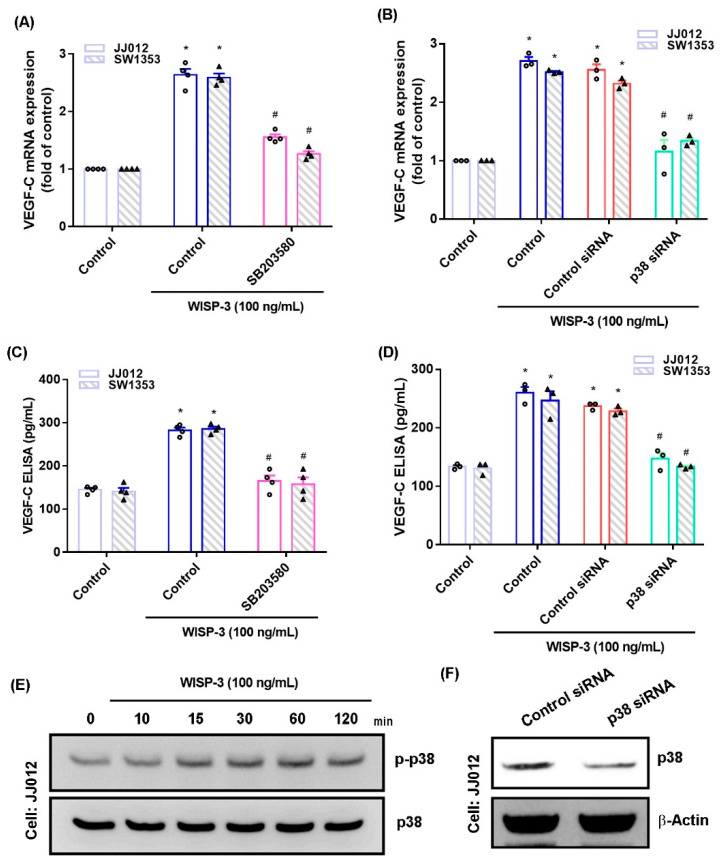
**The p38****pathway controls WISP-3****-induced VEGF-C production**. (**A**–**D**) Cells were pretreated with a p38 inhibitor (SB203580) or transfected with a p38 siRNA, then stimulated with WISP-3. Levels of VEGF-C expression were examined by qPCR and ELISA. (**E**) JJ012 cells were incubated with WISP-3 for the indicated time intervals, and p38 phosphorylation was examined by Western blot. (**F**) JJ012 cells were transfected with p38 siRNA, and p38 expression was examined by Western blot. Quantitative results are expressed as the mean ± SD. All experiments were repeated 3 to 5 times (solid squares, solid dots, hollow circles and solid triangles). * *p* < 0.05 compared with the control group; # *p* < 0.05 compared with the WISP-3-treated group.

**Figure 5 biomedicines-09-01330-f005:**
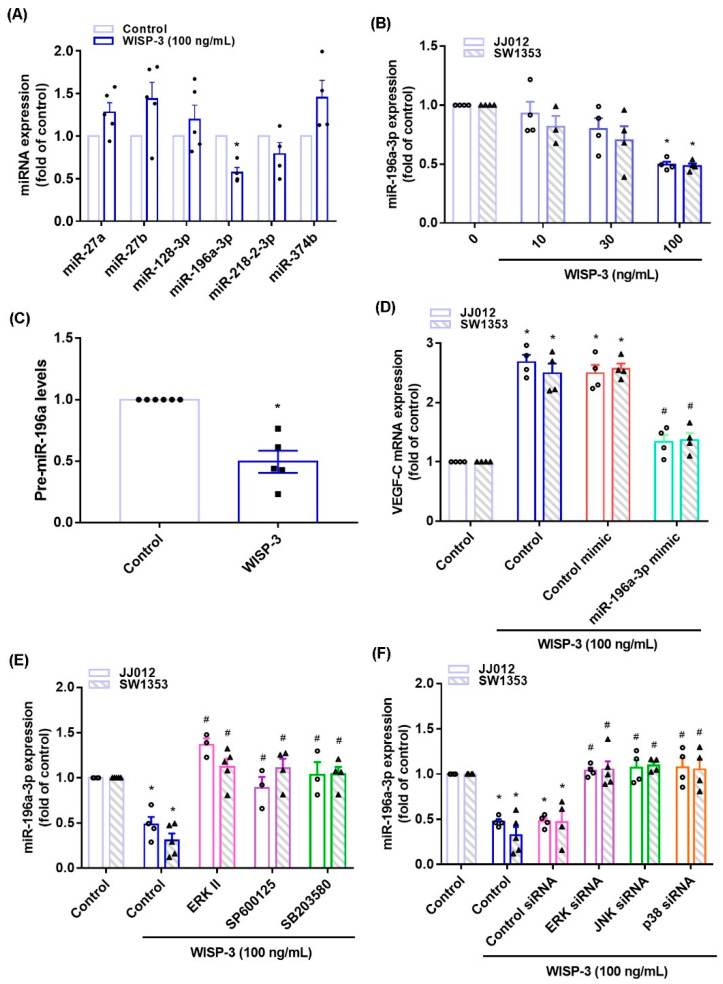
**WISP-3 facilitates VEGF-C synthesis by inhibiting miR-196a-5p.** (**A**) JJ012 cells were incubated with WISP-3. MiRNA expression was determined by the qPCR assay. (**B**,**C**) Cells were incubated with WISP-3. MiR-196a-5p or pre-miR-196a-5p expression was determined by the qPCR assay. (**D**) Cells were transfected with miR-196a-5p mimic, then stimulated with WISP-3. VEGF-C levels were determined by qPCR. (**E**,**F**) Cells were treated with the indicated inhibitors or siRNAs, then stimulated with WISP-3. MiR-196a-5p expression was quantified by qPCR. Quantitative results are expressed as the mean ± SD. All experiments were repeated 3 to 5 times (solid squares, solid dots, hollow circles and solid triangles). * *p* < 0.05 compared with the control group; # *p* < 0.05 compared with the WISP-3-treated group.

**Figure 6 biomedicines-09-01330-f006:**
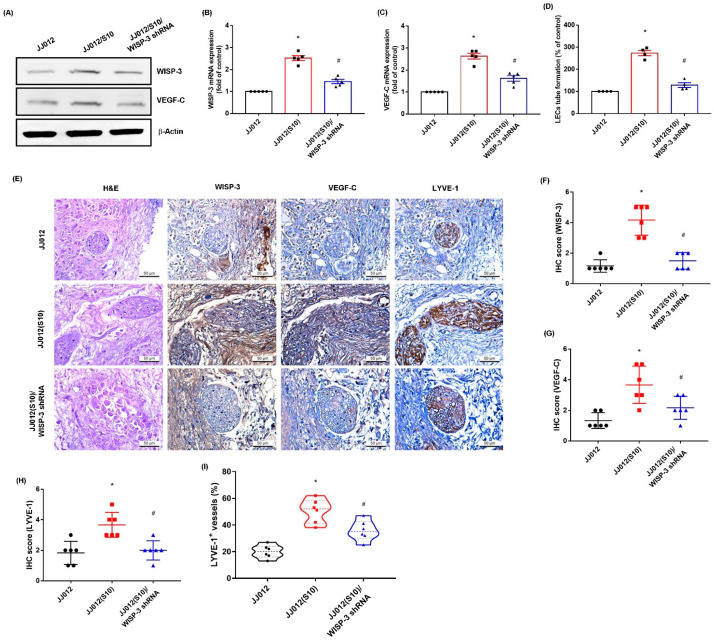
**Inhibition of WISP-3 reduces LEC****lymphangiogenesis in vivo.** (**A**–**C**) WISP-3 and VEGF-C expression were examined by Western blot and qPCR in the indicated cells. (**D**) CM was collected from the indicated cells then applied to LECs and LEC tube formation was measured. (**E**–**I**) At 28 days after the mice were injected, the tumors were embedded in paraffin and sections were immuno-stained using WISP-3, VEGF-C, and LYE-1 antibodies. Quantitative results are expressed as the mean ± SD. All experiments were repeated 3 to 6 times (solid squares, solid dots, hollow circles and solid triangles). * *p* < 0.05 compared with the JJ012 group; # *p* < 0.05 compared with the JJ012(S10) group.

**Figure 7 biomedicines-09-01330-f007:**
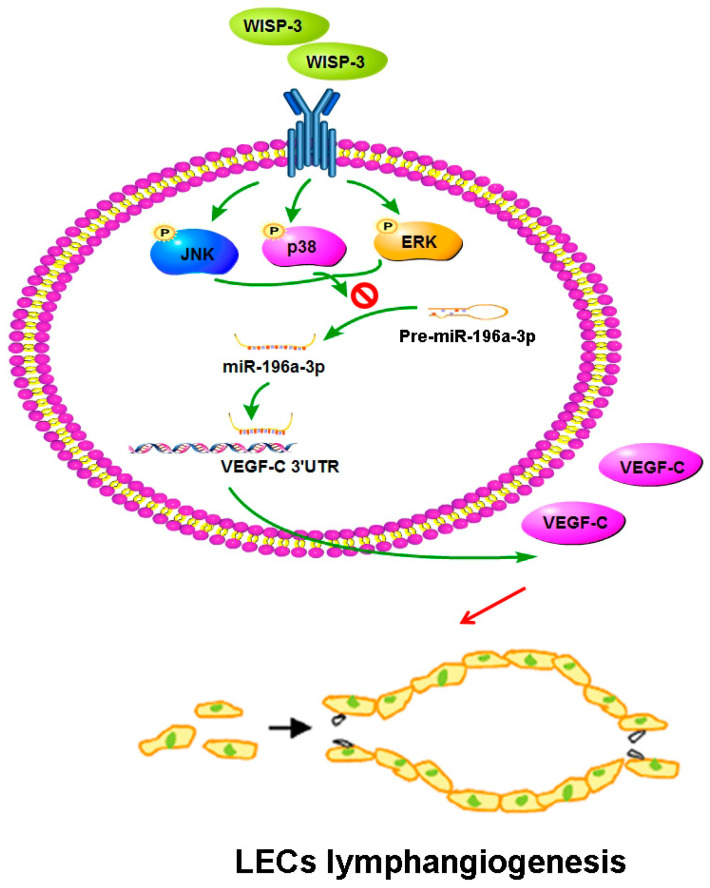
**Schema illustrating the effects of WISP-3 in VEGF-C-dependent lymphangiogenesis in chondrosarcoma.** WISP-3 facilitates VEGF-C production in chondrosarcoma cells and subsequently promotes LEC lymphangiogenesis by inhibiting miR-196a-5p synthesis in the ERK, JNK, and p38 pathways.

## Data Availability

The authors declare that all data supporting the findings of this study are available within the article. The datasets used and/or analyzed during this study are available from the corresponding authors upon reasonable request.

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
