# Peer review of "WISP-3 Stimulates VEGF-C-Dependent Lymphangiogenesis in Human Chondrosarcoma Cells by Inhibiting miR-196a-3p Synthesis"

_biomedicines, 2021, doi:10.3390/biomedicines9101330_

Round 1

Reviewer 1 Report

The study has examined the role of WISP-3 in chondrosarcoma lymphangiogenesis. WISP-3 is already known to play a role in angiogenesis and metastasis associated with chondrosarcoma.

Overall this is a well presented paper that has used a number of specific drugs, antibodies and shRNA to demonstrate the role of WISP-3/VEGF-C and the miRNA, miR-196a-3p in the signalling of chondrosarcoma cells and have attempted to connect this signalling to lymphangiogenesis, which might contribute to the role of these pathways in chondrosarcoma metastasis and tumour progression. The initial in vitro experiments showing the dependency of the signalling pathways for the production of VEGF-C mRNA and protein, and the role in LEC tube formation, and further the modulation of VEGF-C via the JNK and p38 pathways seems straightforward. The data with a mimic to miR-196a-3p shows a link to the pathway, but further in vivo demonstration of the link is not presented.

The major issues I had with the paper relate to the data presented in Figure 6 that relates to the in vivo generation of “lymphangiogenesis” in a tumour model of JJ012 cells and the demonstration of a key finding that the WISP-3 induced VEGF-C leads to actual lymphangiogenesis.

  1. To define these changes as altering lymphangiogenesis or even lymphatic vessel remodelling there needs to be some evaluation of whether new or altered vessels have been formed. Looking at Figure 6 the authors have evaluated the levels of LYVE-1, a marker of smaller lymphatic vessels. It is not evident from the histology/IHC shown in Fig 6E that these structures are lymphatic vessels or that there has been an increase in the number of lymphatic vessels that one might see in the case of lymphangiogenesis. This could be achieved by a reanalysis of the LYVE-1 stained tumours to determine any obvious vessels that are stained and then determine a way to quantify their levels.
  2. Figure legend describes “LEC angiogenesis” which is a confusing use of the two terms. Is this refereeing to lymphnagiogenesis? This needs to be clarified and align with other descriptions that relate to Figure 6.
  3. The Figure 6 legend states that experiments have been performed 3-6 times, does this also hold for the in vivo experiments? Are the statistics derived for each of the sections of Figure 6 from the 3-6 experiments or are they from one representative experiment?
  4. If the authors are to use the term “lymphangiogenesis” they need to demonstrate from the in vivo tumour experiments that they do in fact generate new lymphatic vessels over and above what might be generated from presence of other factors (e.g. VEGF-A).
  5. Are there any parameters of metastasis that can be evaluated in the tumour model? Do these tumours spread to regional lymph nodes to other organs (e.g. lung)?

Author Response

Reviewer 1

  1. To define these changes as altering lymphangiogenesis or even lymphatic vessel remodelling there needs to be some evaluation of whether new or altered vessels have been formed. Looking at Figure 6 the authors have evaluated the levels of LYVE-1, a marker of smaller lymphatic vessels. It is not evident from the histology/IHC shown in Fig 6E that these structures are lymphatic vessels or that there has been an increase in the number of lymphatic vessels that one might see in the case of lymphangiogenesis. This could be achieved by a reanalysis of the LYVE-1 stained tumours to determine any obvious vessels that are stained and then determine a way to quantify their levels.

A: We thank the Reviewer for this suggestion. Our subsequent reanalysis of the LYVE-1 staining identified substantial numbers of lymphatic vessels, supporting the contention that WISP-3 mediates lymphangiogenesis. The reanalysis of LYVE-1 staining is described in the Methods section (2.8 Immunohistochemistry (IHC) staining) and illustrated by Figure 6I. The new text is as follows:

“Intratumoral lymph vessels were defined as LYVE-1-positive (LYVE-1+) vessels that were in close contact with tumor cells or located in the desmoplastic stroma. Peritumoral lymph vessels were defined as LYVE-1+ vessels at a maximum distance of 2 mm from the tumor periphery. In each case, five microscopic fields "lymphatic vessel hot spots" were examined at high power. The mean of these five values was recorded as the LYVE-1+. Density measurements of lymph vessels area were performed with ImageJ software [31,32].” (Lines 153-159)

  1. Figure legend describes “LEC angiogenesis” which is a confusing use of the two terms. Is this refereeing to lymphnagiogenesis? This needs to be clarified and align with other descriptions that relate to Figure 6.

A: We thank the Reviewer for this suggestion. We have corrected the legend for Figure 6 to read as: “Inhibition of WISP-3 reduces LEC lymphangiogenesis in vivo.” (Line 252)

  1. The Figure 6 legend states that experiments have been performed 3-6 times, does this also hold for the in vivo experiments? Are the statistics derived for each of the sections of Figure 6 from the 3-6 experiments or are they from one representative experiment?

A: We thank the Reviewer for this observation. (i) All of our experiments were repeated 3 to 6 times, including the in vivo experiments. (ii) The statistics for each experiment are derived from the repeated experiments, not from one representative experiment. Each graph in Figure 6 depicts how many experiments were performed for each investigation.

  1. If the authors are to use the term “lymphangiogenesis” they need to demonstrate from the in vivo tumour experiments that they do in fact generate new lymphatic vessels over and above what might be generated from presence of other factors (e.g. VEGF-A).

A: We thank the Reviewer for this reminder. Our Discussion now addresses the involvement of lymphangiogenesis in this study, as follows:

“Various lymphangiogenic factors, including VEGF-A, VEGF-C and VEGF-D, are involved in the lymphangiogenic processes of several different diseases, including cancer [9,10,46]. We have previously reported that WISP-3 promotes VEGF-A expression in chondrosarcoma cells and induces endothelial progenitor cell angiogenesis [18]. In addition, an in vivo tumor xenograft study reveals that inhibiting WISP-3 expression reduces the expression of the vessel markers CD31 and VEGF-A, indicating that WISP-3 enhances angiogenesis in vivo [18]. Here, we report that incubation of chondrosarcoma cell lines with WISP-3 concentration-dependently promotes mRNA and VEGF-C synthesis, resulting in LEC lymphangiogenesis. VEGF-C antibody abolished LEC tube formation in CM from WISP-3-treated chondrosarcoma cells. Moreover, we also used a mouse tumor xenograft model to simulate the environment of initial lymphangiogenesis. Our results revealed that knockdown of WISP-3 inhibited chondrosarcoma-promoted expression of the LEC markers VEGF-C and LYVE-1, indicating that VEGF-C is a critical factor in WISP-3-induced lymphangiogenesis in human chondrosarcoma. Whether other lymphangiogenic factors also regulate WISP-3-enhanced promotion of lymphangiogenesis in chondrosarcoma needs further investigation.” (Lines 292-308)

  1. Are there any parameters of metastasis that can be evaluated in the tumour model? Do these tumours spread to regional lymph nodes to other organs (e.g. lung)?

A: We thank the Reviewer for this observation. As our mouse tumor xenograft model was limited to modeling the environment of initial lymphangiogenesis, we were unable to consider the spread of tumors to regional lymph nodes or to metastasis in other organs. Now, our Discussion text clearly indicates that our in vivo model was limited to the initial stage of lymphangiogenesis:

“Moreover, we also used a mouse tumor xenograft model to simulate the environment of initial lymphangiogenesis.” (Lines 302-303)

 Now that all feedback from your Reviewers has been attended to, we sincerely hope that this revised manuscript is suitable for publication in Biomedicines.

Best regards,

Chih-Hsin Tang, PhD.

Reviewer 2 Report

In the present manuscript Lin et al showed that WNT1-inducible signaling pathway protein 3 (WISP-3), increases vascular endothelial growth factor (VEGF)-C synthesis and assisted lymphatic endothelial cells (LEC) lymphangiogenesis in human chondrosarcoma via blocking miR-196-3p in ERK, JNK and p38 signaling pathways. Authors also reported that inhibition of WISP-3 expression decreased VEGF-C dependent lymphangiogenesis in vivo.  Based on these data authors claimed that WISP-3 a potential and novel therapeutic target of chondrosarcoma.  Chondrosarcomas are primary malignant bone tumors that have a poor prognosis and identifying therapeutic target will help in developing novel therapeutic option to treat this disease. This is an interesting study and provide new insight of role of WISP-3 in chondrosarcoma.    
I have one comment to this study. 

Authors showed that WISP-3 mediated VEGF-C mediated lymphangiogenesis in human chondrosarcoma cells by inhibiting miE-196a-3p synthesis. Authors provided data from chondrosarcoma cell lines only, and it would be better if author also used human mesenchymal stromal cells (MSCs) and normal human cartilage cell lines such as CHON-001 and C20A4 in this study.  Data from MSCs and normal human cartilage cells will clearly show that WISP-3 effect is specific to chondrosarcoma cells or not.

Author Response

Assistant Editor, MDPI Belgrade

Biomedicines

Dear Ms. Milosevic,

We greatly appreciate the comments from your Reviewers on our manuscript, WISP-3 stimulates VEGF-C-dependent lymphangiogenesis in human chondrosarcoma cells by inhibiting miR-196a-3p synthesis (biomedicines-1360767). We have carefully revised the manuscript according to their suggestions, using red font to mark up the changes in our Word file. We have addressed their specific points in the text below.

Reviewer 2

  1. Authors showed that WISP-3 mediated VEGF-C mediated lymphangiogenesis in human chondrosarcoma cells by inhibiting miE-196a-3p synthesis. Authors provided data from chondrosarcoma cell lines only, and it would be better if author also used human mesenchymal stromal cells (MSCs) and normal human cartilage cell lines such as CHON-001 and C20A4 in this study.  Data from MSCs and normal human cartilage cells will clearly show that WISP-3 effect is specific to chondrosarcoma cells or not.

A: We thank the Reviewer for this suggestion. Our findings suggest that WISP-3 is important in chondrosarcoma. In addition, we have accordingly addressed the reported effects of WISP-3 in MSCs and in normal human cartilage cells in the Discussion section (see below). Therefore, our research results and the literature suggesting that WISP-3 is important in bone disease.

“WISP-3 protein is involved in the development, homeostasis and repair of mesenchymal tissues [43]. WISP-3 dose-dependently stimulates the migration of undifferentiated mesenchymal stem cells (MSCs), and the chemotactic activity of WISP-3 in these cells is reportedly mediated by integrin ανß5 [43]. WISP-3 also has effects on cartilage homeostasis. For example, WISP-3 overexpression in normal cartilage (C-28/I2) cells dramatically reduces the expression of ADAMTS-4 and ADAMTS-5, and markedly elevates matrix metalloproteinase-1 (MMP-1) and MMP-10 expression, leading to the degradation of cartilage and development of osteoarthritis [44]. Thus, WISP-3 plays a critical role in influencing the development of bone disease, and by the recruitment of MSCs by stimulating their migration in skeletal development and repair.” (Lines 277-286)

    Now that all feedback from your Reviewers has been attended to, we sincerely hope that this revised manuscript is suitable for publication in Biomedicines.

Best regards,

Chih-Hsin Tang, PhD.

Round 2

Reviewer 1 Report

The authors have addressed the questions I raised about the paper in full.